# Influence of Benzothiadiazole on the Accumulation and Metabolism of C6 Compounds in Cabernet Gernischt Grapes (*Vitis vinifera* L.)

**DOI:** 10.3390/foods12193710

**Published:** 2023-10-09

**Authors:** Jianfeng Wang, Yuqi Han, Chunxia Chen, Faisal Eudes Sam, Ruwen Guan, Kai Wang, Yu Zhang, Man Zhao, Changxia Chen, Xuan Liu, Yumei Jiang

**Affiliations:** 1Gansu Key Laboratory of Viticulture and Enology, Gansu Wine Industry Technology R&D Center, College of Food Science and Engineering, Gansu Agricultural University, Lanzhou 730070, China; wjf971112@163.com (J.W.);; 2College of Enology, Northwest A&F University, Xianyang 712100, China

**Keywords:** grapes, Cabernet Gernischt, C6 compounds, benzothiadiazole, pre-swelling stage

## Abstract

Pre-harvest application of elicitors improves grape quality, specifically the phenolic compounds and color characteristics. Limited research has been conducted on the impact of elicitors on the C6 compounds found in grapes. This is due to lack of comprehensive studies examining the combined effects of bound aroma compounds, enzyme activity, and substrate availability. This study aimed to assess the impact of benzothiadiazole (BTH) on the physicochemical properties and C6 compounds of Cabernet Gernischt grapes during ripening. Compared with the control group (CK), BTH treatment significantly increased the 100-berry weight, skin/berry ratio, pH, total phenolic content, and total flavonoid content in ripe grapes. Additionally, BTH treatment led to significant reductions in reducing sugar, total soluble solids, titratable acidity, linoleic acid, linolenic acid, and free C6 aldehydes. Furthermore, BTH treatment significantly decreased the contents of free C6 alcohols and increased the levels of free and bound C6 esters. BTH treatment also increased the activities of lipoxygenase, alcohol dehydrogenase, and alcohol acetyltransferase enzymes, while it decreased the activity of hydroperoxide lyase enzyme. The application of BTH resulted in changes to the physicochemical properties and levels of C6 compounds in Cabernet Gernischt grapes by up-regulating enzyme activity and down-regulating precursors.

## 1. Introduction

The aroma of grapes and wine is an important quality indicator that influences consumer preferences. It is influenced by various factors, including grape variety, vintage, climate, cultivation practices, and ripeness [1,2]. Grape aroma compounds primarily originate from the metabolic conversion of amino acids, sugars, and fatty acids via the amino acid, isoprene, and fatty acid pathways. Among them, C6 compounds, such as C6 aldehydes, C6 alcohols, and C6 esters, are derived from the fatty acid pathway. These compounds are produced from linolenic and linoleic acids, which are catalyzed by hydroperoxide lyase (HPL), lipoxygenase (LOX), alcohol acetyltransferase (AAT), and alcohol dehydrogenase (ADH) [1]. C6 compounds contribute to the aromatic profile of grape berries and wines, imparting “green leafy”, “grassy”, or “fresh plant” odors. Additionally, these compounds serve as precursors for hexyl acetate in wines and the distinctive aroma compound 3-mercapto-1-hexanol (3MH) found in New Zealand’s Sauvignon Blanc white wine [3,4]. Several factors have been found to influence the production and accumulation of C6 compounds in grape cultivation. These factors include the viticultural environment climate [5,6], grape variety [7,8], maturity stages [8,9], cluster thinning [10,11], light condition of the berry microclimate [11], training system [12,13], rainfall [14], and exogenous spraying [15,16]. Moreover, a recent study was conducted to examine the influence of the grape ripening stage and phytosanitary state on C6 compounds, revealing significant effects [17]. Among these factors, exogenous spraying of elicitor is considered a green, environmentally friendly, highly efficient, stabilized, and non-residual cultivation management strategy that is of great interest in grape cultivation.

Benzothiadiazole (BTH) is a commercial chemical elicitor that activates systemic acquired resistance (SAR) in plants by mimicking the endogenous signaling molecule salicylic acid (SA). In grapes, BTH induces the plant’s natural immune response; triggers local or systemic defense responses; and contributes to the increase in cell wall proteins, phenolics, and cellulose composition [18,19], resulting in an enhanced color intensity [20,21]. Moreover, it has been observed that BTH exhibits a diminishing effect on the levels of essential metabolites, particularly soluble sugars and amino acids, in grapes [22,23]. Additionally, it has been found to modify the composition of soluble sugars [23]. Soluble sugars have a dual role in fruits. They not only affect the organoleptic quality of fruits but also serve as important signaling molecules. In grapes, these sugars play a crucial role in influencing the production of secondary metabolites by participating in defense responses and regulating the transcription of metabolic genes [23,24]. In addition, the effect of BTH on grapes’ soluble sugars and total acids alters grapes’ ripening time [19]. BTH affects the grape anthocyanin composition, grape skin tannin, and flavonol content [25,26], as well as the concentration of terpenoids and norisoprenoids in grapes [22]. Previous research has mostly examined the impact of BTH application on the grape cell wall composition [18,19], phenolic compounds [15,21,27], color [19,28], and hydrogen peroxide and reactive oxygen species (ROS) metabolism [18,29,30]. However, a limited number of studies have examined the impact of BTH on the composition of grape aroma compounds [31]. Our previous study found that BTH sprayed on Cabernet Gernischt grapes at the swelling stage resulted in higher concentrations of aroma compounds than BTH-treated grapes at the fruit-set and veraison stages [22]. Therefore, in this study, to further clarify the effects of BTH treatment on the physicochemical parameters and accumulation of C6 compounds in Cabernet Gernischt grapes, an experiment was conducted by spraying 0.37 mM BTH on Cabernet Gernischt grapes at the pre-swelling stage. Subsequently, the physicochemical properties and the composition and concentration of free and bound C6 compounds were analyzed during the ripening process of the grapes. Additionally, the enzyme activities of LOX, HPL, ADH, and AAT were monitored. This aimed to investigate the underlying mechanism behind the effects of BTH treatment on the accumulation and metabolism of C6 compounds in Cabernet Gernischt grapes. The findings of this research could provide a theoretical basis and scientific data for the appropriate application of BTH in the cultivation of Cabernet Gernischt grapes.

## 2. Materials and Methods

### 2.1. Plant Material and Study Site

During the 2022 growing season, a field study was carried out at the experimental vineyard of the Food Science and Engineering College in Gansu Agricultural University, located in Lanzhou, China, at 103°41′ E, 36°5′ N and 1530 m above sea level. The study site is situated in the Hexi Corridor, which offers optimal conditions for cultivating wine grapes due to its exceptional combination of water, soil, sunshine, and diurnal temperature differences. In particular, the site is characterized by a temperate continental climate and experiences an average annual high temperature of 19.0 °C and an average low temperature of 5 °C. The total annual rainfall measures 429.9 mm, while the site benefits from more than 2358 h of sunshine annually. The average wind speed is recorded at 6.9 km/h. The frost-free period extends for 182 days, and there are 48 days of precipitation throughout the year. The main grape varieties grown in this region include Cabernet Sauvignon, Chardonnay, Riesling, Syrah, and Cabernet Gernischt. The experiment was conducted on 13-year-old Cabernet Gernischt vines grown on a vertical trellis system with a row spacing of 2.5 m and a plant spacing of 0.80 m in an east–west orientation. The vineyard was managed in accordance with established standard viticultural practices, such as pruning and shade control, and was irrigated using a drip irrigation system. Two types of fertilizers, urea (46% nitrogen) and ammonium phosphate dibasic (18% nitrogen and 46% phosphorus), were applied to the vineyard in mid-April each year at a rate of 0.75 t/ha. Also, a sheep-based organic fertilizer (organic matter > 45% and N + P_2_O_5_ + K_2_O > 5%) was applied to the vineyard in mid-October each year at a rate of 9 t/ha.

### 2.2. Reagents and Chemicals

Unless otherwise stated, all the reagents used in this study were analytically pure. The water used for various experiments was purified with a Milli-Q purification system. Tween 80, glucose anhydrous, cupric sulfate, potassium sodium tartrate tetrahydrate, sodium hydroxide, magnesium chloride, acetyl-CoA, linoleic acid sodium salt, nicotinamide adenine dinucleotide, dithiothreitol, Triton X-100, 4-morpholineethanesulfonic acid, polyvinylpolypyrrolidone (PVPP), dipotassium hydrogen phosphate, potassium dihydrogen phosphate, sodium acetate, acetaldehyde, and butanol were obtained from Yuanye Co., Ltd. (Shanghai, China). BTH, E-2-hexenal, hexanal, Z-3-hexenal, E-2-hexenol, Z-3-hexenol, hexanol, Z-3-hexenyl acetate, hexyl acetate, 2-octanol, linolenic acid, and linoleic acid were HPLC grade and obtained from Sigma-Aldrich (Shanghai, China).

### 2.3. Treatment and Sampling

The treatment method described by Salifu et al. [22] was adopted with minor changes. In this experiment, the CK and BTH treatments were each administered in the form of aqueous solutions to 15 blocks of vines organized in a randomized design. Each block had three biological replications. To prevent contamination, two untreated rows were created as a protective barrier between the treatments. As described in the methodology of Miliordos et al. [21] and Salifu et al. [22], the BTH treatment contained 0.37 mM BTH with Tween 80 as a wetting agent, whereas the control treatment had only Tween 80. The solutions were sprayed with a high-pressure sprayer onto the berry surfaces at the pre-swelling stage. At this stage, the berries had a transverse diameter of 4.0 mm and a vertical diameter of 4.3 mm, approximately 3 weeks post-flowering. No chemicals other than BTH were applied to the experimental grapevines. Berry samples were collected at biweekly intervals from the pre-swelling stage to harvest, which covered the period from 17 June to 14 September. All experiments were conducted using grapes that were phytosanitary and 100% healthy. Several bunches of berries were randomly harvested for each treated sample, bagged, and conveyed to the laboratory. The berries were destemmed and wrapped in aluminum foil in small amounts before being frozen in liquid nitrogen and stored at −80 °C until use. Appendix A contains additional sampling information as well as the determination of the grape ripening stages.

### 2.4. Determination of Physicochemical Properties

First, a beaker was weighed, then it was filled with 100 randomly selected berries and weighed again. The weight of the berries was expressed in g/100 berries. The skin/berry ratio was calculated by dividing the weight of the grape skin by the total weight of the berry. The berries were deseeded, manually pressed, and centrifuged (H2050R, Hunan Xiangyi Laboratory Instrument Development Co., Ltd., Changsha, China) at 8000 r/min for 10 min to obtain a clear juice. The juice was then analyzed for total soluble solids (°Brix) using a PAL-1 pocket refractometer (Atago, Tokyo, Japan) and for pH using a pH meter (Inesa, Shanghai, China). Titratable acidity and reducing sugars were determined according to the method of Salifu et al. [22]. Total phenolics and total flavonoids were determined as described by Gómez-Plaza et al. [26] and Cao et al. [32].

### 2.5. Enzyme Activity

Grape berries were deseeded and ground to powder with liquid nitrogen for the determination of enzyme activities. The activities of enzymes, such as LOX, HPL, ADH, and AAT, were assessed using the method described by Gong et al. [33] with slight modifications. A 0.01 change in absorbance per minute per gram was defined as one enzyme activity unit (U), and LOX, ADH, and AAT activity was expressed as U/g fresh weight (FW).

LOX: Sample powder (5.0 g) was mixed with 6 mL of pre-cooled 0.1 M phosphate buffer containing 1% Triton X-100, 0.35 M PVPP, and a pH of 6.8. The mixture was subjected to a temperature of −18 °C for 5 min. Following this, it was centrifuged at 4 °C, 10,000 r/min, and for 30 min. The resulting supernatant was collected as the crude enzyme solution. The reaction system comprised 25 μL of 10 mM sodium linoleate solution, 2.9 mL of 0.1 M phosphate buffer (pH 6.8), and 75 μL of crude enzyme solution. Using distilled water as a reference, the absorbance was measured at 234 nm. The timing commenced 30 s after the addition of crude enzyme solution and absorbance values were recorded at 15 s intervals for a total duration of 120 s.

HPL: Sample powder (3.0 g) was added to 9 mL of 50 mM sodium acetate buffer (pH 5.0, 0.21% ascorbic acid, 0.5% Triton X-100). Subsequently, the mixture was cooled at −18 °C for 5 min then centrifuged for 30 min at 10,000 r/min and at 4 °C. The supernatant was collected and measured using an enzyme-linked immunosorbent assay kit (Jiwei Biological Technology Co., Ltd., Shanghai, China).

ADH: Sample powder (3.0 g) was added to 9 mL of 0.1 M 4-morpholineethanesulfonic acid buffer (2 mM dithiothreitol, 1% PVPP, pH 6.5), cooled at −18 °C for 5 min, and centrifuged at 4 °C for 30 min at 10,000 r/min. The supernatant was then taken as the crude enzyme solution. The reaction system included 2.0 mL of 0.1 M 4-morpholineethanesulfonic acid buffer, 100 μL of 1.6 mM nicotinamide adenine dinucleotide solution, 200 μL of 80 mM acetaldehyde solution, and 0.2 mL of crude enzyme solution. Using distilled water as a reference, the absorbance was measured at 340 nm. Timing commenced 15 s after the addition of crude enzyme solution and the absorbance was measured every 30 s within 300 s.

AAT: Sample powder (3.0 g) was mixed with 5 mL of 0.5 M Tris-HCL (pH 8.0, 0.1% Triton X-100, 0.3 mg PVPP), extracted in an ice bath for 20 min, and centrifuged at 12,000× *g* r/min for 20 min. The supernatant was collected and used as the crude enzyme solution. The reaction system included 2.5 mL of 5 mM MgCl_2_ solution, 150 μL of acetyl-CoA (0.5 M Tris-HCL, pH 8.0), 50 μL of 200 mM butanol solution (0.5 M Tris-HCL, pH 8.0), and 150 μL of crude enzyme solution. After 15 min in a 35 °C water bath with crude enzyme solution, 100 μL of 10 mM 5,5′-Dithiobis-(2-nitrobenzoic acid) was added. Using reaction solution without crude enzyme solution as a reference, the absorbance was measured at 412 nm. Timing began 15 s after the addition of crude enzyme solution, and absorbance values were recorded every 15 s for 120 s.

### 2.6. Sample Preparation for Aroma Compounds Analysis

The sampled grapes (50.0 g) from each treatment were deseeded and then mixed with 1.0 g of PVPP and 0.5 g of gluconolactone. Subsequently, the mixture was ground to a powder with liquid nitrogen.

Free aroma: Approximately 15.0 g of the powder sample was allowed to stand at 4 °C for 1 h. Following this, a grape mash was prepared for subsequent analysis.

Bound aroma: The extraction method used was based on the method described by Yue et al. [16] with slight modifications. The powder sample was allowed to stand at 4 °C for 2 h and centrifuged at 12,000× *g* r/min for 15 min. Subsequently, 2 mL of the supernatant was passed through an activated solid-phase extraction column. About 2 mL of water and 5 mL of dichloromethane were then added to remove the saccharic acid and free aroma, and 20 mL of methanol was used to elute the bound aroma into a 50 mL round-bottom flask. The eluted mixture was then dried using a rotary evaporator at 30 °C and 80 r/min. Following that, 10 mL of citric/phosphoric acid buffer (0.2 M, pH = 5) and 0.1 g of glycosidase AR2000 were added, and the mixture was placed in a constant temperature water bath at 37 °C for 16 h.

### 2.7. Analysis of Grape Aroma Compounds

Headspace solid-phase microextraction gas chromatography–mass spectrometry (HS-SPME-GC-MS) was used to analyze grape aroma compounds, with modifications to the method reported by Salifu et al. [22]. A 20 mL vial containing 5 g of slurry and 1 g of sodium chloride was weighed. Following that, a small magnetic stir bar and 10 µL of internal standard (50 ppm, 2-octanol) were added. The vial was firmly sealed and immersed in a 40 °C water bath with agitation at 40 r/min for 30 min. The volatile aroma in the vial’s headspace was extracted using a 50/30 µm DVB/CAR/PDMS fiber, which was then thermally desorbed in the injector port of the GC-MS for 10 min.

A gas chromatography–mass spectrometer system (TRACE 1310-ISQ, Thermo Fisher Scientific, San Jose, CA, USA) connected with a DB-WAX column (60 m × 2.5 mm × 0.25 µm, Agilent Technology, Santa Clara, CA, USA) was used to analyze the volatiles. With a flow rate of 1 mL/min, helium gas was employed as the carrier gas. The operating temperature of the injector was configured to 230 °C for splitless injection. The GC temperature program commenced at 50 °C for a duration of 10 min. Subsequently, the temperature was increased at a rate of 3 °C/min until it reached a final temperature of 180 °C, which was maintained for 6 min. The ion source and transfer line temperatures were, respectively, set at 250 and 180 °C. The operating mass range spanned from 50 *m*/*z* to 350 *m*/*z* in a full scan mode with an electron energy of 70 eV. The standard solution was diluted into a variety of solutions with a concentration gradient, and the standard curve for C6 compounds was established using the same assay method. The aroma concentration was calculated based on the peak area of samples and standard equations (Appendix A).

### 2.8. Analysis of Linolenic and Linoleic Acids

The method of determining linolenic and linoleic acids by Ju et al. [34] was adopted with slight modifications. Approximately 6 g of grape powder was added to 25 mL of hexane subjected to ultrasonication at 720 W for 30 min. The mixture was then macerated and left to extract at 4 °C for 24 h. The supernatant was evaporated to dryness using a vacuum rotary evaporator. Then, 5 mL of 1% sulfuric acid/methanol solution was added, and the mixture was methylated in a water bath at 65 °C for 30 min. The supernatant was cooled to 25 °C and mixed with 3 mL hexane. Following that, the supernatant was filtered with a 0.22 μm organic membrane. The measurement was performed using a PerkinElmer Clarus500 gas chromatograph (Agilent HP-88 capillary column, 100 m × 0.25 mm × 0.2 μm) equipped with a flame ionization detector and autosampler. The inlet and detector temperatures were set at 250 °C and 300 °C, respectively. The programmed temperature was set as follows: 120 °C for 1 min, followed by a gradual increase of 10 °C/min to reach 175 °C (held for 10 min). Then, the temperature was increased at a rate of 5 °C/min to attain 210 °C (held for 5 min) and finally increased at a rate of 5 °C/min to reach 230 °C (held for 25 min). Nitrogen was used as the carrier gas at a flow rate of 1 mL/min. Quantification was conducted using the external standards listed in Appendix A.

### 2.9. Statistical Analysis

The data were analyzed in triplicate and the results were reported as means. IBM SPSS statistical 26.0 for Windows (SPSS Inc., Chicago, IL, USA) was employed for data analysis. One-way ANOVAs were performed on all parameters to compare the differences in different groups and the differences at different weeks post-flowering. The means were compared using the post hoc Tukey’s test, and differences at *p* < 0.05 were considered significant. Additionally, Pearson correlation analysis was performed on fatty acids, enzyme activities, and C6 compounds using OmicStudio tools available at https://www.omicstudio.cn, accessed on 4 September 2022. Metaboanalyst 5.0 (https://genap.metaboanalyst.ca, accessed on 5 September 2022) was also used to generate hierarchical heatmaps of grape aroma compounds and Orthogonal Projections to Latent Structures Discriminant Analysis (OPLS-DA).

## 3. Results and Discussion

### 3.1. Influence of BTH Treatment on the Physicochemical Parameters of Cabernet Gernischt Grapes

#### 3.1.1. 100-Berry Weight and Skin/Berry Ratio

Cabernet Gernischt grapes underwent two rapid expansions during grape ripening, resulting in an increase in the weight of 100 berries. Specifically, the weight increased by 48.53% and 40.83% at the end-swelling stage (7–9 weeks) and veraison (9–11 weeks), respectively. However, the rate of increase slowed down as the grapes approached full ripening (13–15 weeks). The BTH treatment had a significant effect on the weight of 100 berries, particularly the ripe berries in week 15. The weight of ripe berries (22.79 g) significantly increased by 16.33% compared to the CK (Figure 1A). The weight of grape berries is influenced by factors such as vintage, precipitation, and temperature. Previous studies on the application of BTH to Syrah and Monastrell grapes have demonstrated a significant increase in grape berry weight, which was attributed higher levels of precipitation [19,25,28]. In contrast, Salifu et al. [22] found that BTH accelerated the rate of absorption of water molecules in grapes. However, if BTH only affects the rate of absorption of water molecules by grapes without the acceleration of cell division and accumulation of primary metabolites within the grapes, no significant difference in weight would be observed between BTH-treated and CK grapes, precisely because we applied BTH during the first phase of rapid growth and development of the grapes. So, this research finding suggests that BTH treatment might increase the berry weight by enhancing cell division, expansion, and metabolite accumulation in grapes, in combination with the analysis of grape phenology and developmental patterns [35].

BTH treatment had no significant effect on the skin/berry ratio during grape ripening. However, ripe grapes showed an increase of 11.47% compared to the CK (Figure 1B). The rapid expansion of grapes at the initial swelling stage caused a decrease and subsequent increase in the skin/berry ratio before veraison. This trend continued after veraison and until harvest, possibly due to increased skin contents during the second rapid expansion and ripening of the grapes. The weight and skin/berry ratio of grapes can significantly influence the quality of their resultant wine. Thus, the compositional ratios of phenolics, anthocyanins, and tannins in the skin, flesh, and seeds of berries affect leaching and diffusion during wine fermentation. BTH can enhance the accumulation of lignin and its hydroxyproline-rich glycoprotein in plant cell walls, leading to the thickening of cell walls or the formation of papillae. Additionally, BTH promotes the deposition of calcium ions and induces the accumulation of callose at invasive sites [36]. Furthermore, BTH is an SA analog and SA has been found to delay the decline in protopectin and cellulose content, as well as the increase in water-soluble pectin content in grapes. It also delays the rise in activity of cell wall-degrading enzymes, such as polygalacturonase, pectin methylesterase, β-galactosidase, β-glucosidase, and cellulase. Additionally, SA inhibits the expression of *VvPG*, *VvPME*, *Vvβ-Gal*, and *Vvβ-Glu*, thereby strengthening the skin cell wall [37]. BTH treatment significantly increased the cell wall protein and cellulose content of Monastrell grape berries but showed variety dependence [19]. The increase in cellulose and the thickening of grape skins may affect the efficiency of anthocyanin extraction during winemaking. From the results (Figure 1B), it is evident that BTH, an SA analog, can alter the grape skin/berry ratio of grapes by modulating the composition of the grape skin cell wall.

#### 3.1.2. Total Soluble Solids and Reducing Sugars

Total soluble solids (TSSs) not only affect the post-harvest flavor and overall sensory quality of grapes but also play a crucial role as signaling molecules in regulating gene transcription related to defense responses, metabolic processes, grape ripening, and the biosynthesis of secondary metabolites [24]. In this study, the levels of reducing sugars in grapes exhibited a consistent increase during the ripening process, while TSSs initially decreased and then subsequently increased. The TSS content increased rapidly during veraison, with the highest content in ripe grapes. BTH treatment resulted in a significant decrease in reducing sugars and TSSs in ripe grapes. Specifically, at the mid-ripening stage (week 13), reducing sugars and TSSs were reduced by 16.50% (31.73 g/L) and 11.13% (2.30 °Brix), respectively, compared with the CK (Figure 1C,D).

Li et al. [30] found that BTH treatment reduces the activities of sucrose synthesis enzymes (such as sucrose synthase and sucrose phosphate synthase) and the expression of key genes involved in sucrose synthesis (*VvSuS*, *VvSPS*). Conversely, BTH treatment increases the activities of sucrose invertase enzymes (e.g., cell wall-bound acid invertase, soluble acidic invertase, neutral invertase) and their relative gene expression levels. As a result, the sucrose content decreases, while the fructose and glucose contents increase [23]. Glucose can also be synthesized and catalyzed by glucose phosphoglucomutase to generate UDP-glucose and ADP-glucose, which serve as precursors for the synthesis of phenolics and starch, respectively [15]. Therefore, the results indicate that BTH promoted glucose metabolism by increasing phosphoglucomutase activity, resulting in a decreased glucose content and increased phenolic content. Genetic research has demonstrated significant associations between sugar and plant hormone signaling, with hexokinase (HXK) playing a crucial role as a glucose sensor [24]. Various signals from sugar activate multiple pathways, some of which depend on HXK, while others do not, and these pathways use different mechanisms to regulate transcription, translation, protein stability, and enzymatic activity [24]. HXK, phosphofructokinase, and pyruvate kinase (PK) are the three most important pacemaker enzymes in the energetic EMP pathway. In plants, HXK is responsible for catalyzing the metabolism of different hexoses, including fructose as one of its substrates. Phosphofructokinase is involved in the phosphorylation of fructose 6-phosphate, resulting in the formation of 1,6 fructose diphosphate. Pyruvate is a significant product of PK and the final product of the EMP pathway. It serves as a crucial precursor for the synthesis of organic acids and plays a role in various metabolic pathways, including sugar metabolism, fatty acid metabolism, and the tricarboxylic acid cycle. Additionally, pyruvate is involved in the respiratory oxidative energy supply and is closely associated with several important metabolic processes [24]. From our findings, with fructose and pyruvate as precursors, BTH could potentially regulate the synthesis of aroma substances by reducing the expression of *VvHXK* and *VvPK* genes; inhibiting the activities of HXK and PK enzymes; and altering the levels of fructose (increasing) and pyruvate (decreasing). Similar conclusions were reached in a study where SA was applied to peaches [38]. The ripening process of BTH-treated grapes was delayed, as evidenced by the reduction in TSSs and reducing sugars. This finding aligns with a previous study that investigated the effects of BTH treatments on Monastrell, Merlot, and Cabernet Sauvignon grapes. In that study, this decrease was attributed to higher rainfall. Moreover, the effect of BTH is likely to be influenced by meteorological conditions, as the greatest TSSs reduction in BTH-treated grapes occurred in a vintage with higher rainfall [19].

#### 3.1.3. Titratable Acidity and pH

The titratable acidity (TA) of grapes initially increased and then decreased during the ripening process. The highest content of TA was observed at the mid-swelling stage (week 5), after which there was a rapid decline at veraison. The TA content of BTH-treated grapes was significantly lower than that of CK grapes at end-veraison (week 11) and when the grapes were ripe (week 15), with reductions of 1.49 g/L and 0.77 g/L, respectively (Figure 1E). The pH of grapes also increased and decreased during the ripening process. It reached its highest peak at mid-ripening (week 13) and then declined to a level similar to that observed at veraison. The pH of BTH-treated grapes significantly increased by 5.48%, 1.97%, and 3.35% compared to the CK at the stages of end-veraison, mid-ripening, and ripe grapes, respectively (Figure 1F). In summary, BTH treatment reduced TA and increased the pH during grape ripening. The results are consistent with a previous study that treated Muscat Hamburg with 200 mg/L SA [16].

Cabernet Gernischt grapes contain organic acids, such as tartaric, malic, and citric acids [39]. Citric acid is involved in the synthesis of fatty acid, which is a precursor to suberin polyaliphatic, and ATP-citrate lyase hydrolyzes citric acid in the plastids, producing oxaloacetate and acetyl-CoA [40]. Fatty acid synthesis is limited to plastids due to the inability of acetyl-CoA to traverse mitochondrial membranes [41]. Therefore, citric acid synthesized in the mitochondria can be transported to the cytosol and subsequently to the plastids [42]. Studies have shown that BTH facilitates the deposition of suberin polyaliphatic in wounds [40]. BTH treatment of Monastrell grapes at the veraison stage did not result in statistically significant differences in total acidity and tartaric acid content; however, malic acid concentrations were lower in the treated grapes than in control grapes [25]. The mechanism hypothesized in our study is similar to that observed in a previous study where BTH was found to decrease citric acid levels during wound healing in muskmelons [40]. Therefore, it can be hypothesized that BTH mainly affects the citric and malic acid content during grape ripening, although this effect is subject to variations based on grape variety.

#### 3.1.4. Total Phenolic (TP) and Total Flavonoid (TF) Content

The TP content of grapes decreased progressively during ripening. The decline was greater at the end-swelling stage (weeks 7–9), while the trend was moderate from end-veraison to ripening (weeks 11–15). These results were similar to reported by Cao et al. [32]. BTH application significantly increased the TP content during specific weeks after veraison (weeks 9, 11, and 15). Notably, at end-veraison (week 11), the TP content of BTH-treated grapes significantly increased by 30.78% compared to the CK (Figure 1G). Flavonoids are significant plant secondary metabolites that serve various functions, including attracting pollinators, providing UV protection, acting as antioxidants, and aiding in the plant defense. The content of TF decreased during the ripening process of grapes. However, ripe grapes treated with BTH showed a significant increase of 17.23% compared to the CK (Figure 1H).

The main components of the grape TP content are hydroxycinnamic acid derivatives, flavanols (including proanthocyanidins), flavonols, and anthocyanins. The levels of flavonols, proanthocyanidins, and hydroxycinnamic acids are higher during the fruit-set stage and decrease significantly during the ripening stage [43]. In this study, the decrease in these compounds may be the primary factor contributing to the decline in the TP and TF content as the grapes matured. The stress response caused by BTH triggers the activation of secondary metabolic pathways, resulting in the accumulation of phenolics. Cao et al. [32] found that BTH effectively increased strawberry phenylalanineammonialyase activity and elevated the levels of total phenolics, flavonoids, and anthocyanins. Previous studies have also proposed that BTH enhances chalcone synthase activity in the process of polyphenol and anthocyanin biosynthesis [44]. Additionally, it has been found that BTH suppresses the activity and gene expression of phospholipase enzymes, such as phospholipase A_2_, phospholipase C (PLC), and phospholipase D (PLD) [45]. Furthermore, the exogenous application of BTH enhances the expression of the MYB-bHLH-WD40 complex, which is related to proanthocyanidins, and increases the content of flavan-3-ol, which leads to the accumulation of proanthocyanidins [46]. Treatment with exogenous SA activated the activity of flavonol synthase (FLS) and stimulated the synthesis of new FLS proteins at a specific stage [47]. SA may regulate the biosynthesis of grape flavonols by increasing the activity of FLS. The accumulation of flavonols may also play a role in the formation of SA-mediated acquired resistance [47]. Moreover, SA promoted the accumulation of TP, TF, total flavan-3-ols, and flavan-3-ol monomers in grapevine leaves. This led to an increase in the activities of anthocyanin reductase and leucoanthocyanidin reductase, as well as the transcription of genes related to flavan-3-ol biosynthesis (e.g., *VvANR*, *VvLAR1*, *VvLAR2*, *VvMYBPA1*, and *VvMYBPA2*) [48]. Therefore, BTH may enhance the accumulation of proanthocyanidins, flavonols, and flavanols, as well as increase the overall phenolic content through the regulation of grape chalcone synthase, phospholipase, FLS, anthocyanin reductase, and leucoanthocyanidin reductase activities, as well as the expression of related genes [32,43,44,45,46,47,48]. Overall, BTH positively affected the anthocyanin content of Monastrell [25] and Merlot [49] grapes. The activation of enzymes related to anthocyanin metabolism [26], enzymes in the phenylpropanoid pathway [50], and enzymes during polyphenol synthesis [32,44] by BTH has been widely reported. Based on our research findings, it is easy to conclude that BTH increases the phenolics content mainly by modulating the activity of enzymes involved in metabolic pathways and may not be limited by grape variety.

### 3.2. Effect of BTH on Linolenic and Linoleic Acids in Cabernet Gernischt Grapes

Linolenic and linoleic acids are the direct precursors of the LOX pathway. Linolenic and linoleic acids increased and then decreased during grape ripening. Linoleic acid was 1.75–11.02 times greater than linolenic acid, and the difference between the two was most pronounced during the mid-ripening stage at week 13 (Figure 2B). This is consistent with the results reported in the study by Ju et al. [34]. BTH treatment resulted in an increased accumulation of linoleic and linolenic acid before veraison, while the opposite effect was observed after veraison (Figure 2). It can be hypothesized that BTH had a promoting effect on acetyl-CoA before veraison [51], which enhanced the conversion of bound fatty acids to free linolenic and linoleic acids. Fatty acid desaturase activity was found to decrease in BTH-treated grapes after veraison, whereas the conversion of bound to free was reduced [45]. In addition, BTH treatment inhibited the relative expression of phospholipid metabolism PLC and PLD genes, leading to a decrease in PLC and PLD activities [45]. Consequently, this reduction in enzymatic activity results in a decrease in the production of phospholipid metabolites, such as linolenic acid and linoleic acid. Overall, the erratic trend of linolenic and linoleic acid concentrations in BTH-treated grapes before and after veraison may be the result of a combination of the factors mentioned above.

The content and proportion of unsaturated fatty acids affect cell membrane fluidity, which has an important impact on physiological responses, such as resistance and senescence in grapes [45]. In our study, the unsaturation of fatty acids in BTH-treated grapes increased during grape ripening, which is in line with the results of a previous study that reported that BTH treatment can inhibit the degradation of phospholipid and maintain the fluidity of the cell membranes [45]. Drought stress causes an increase in triphosphopyridine nucleotide oxidase activity at the plasma membrane and the accumulation of a large number of ROS, which when exceeding the threshold will attack the cell membrane system and cause membrane lipid peroxidation. This process involves breaking the double bonds of unsaturated fatty acids, leading to the formation of lipohydroperoxides, which ultimately degrade into malondialdehyde [52]. However, BTH increases the accumulation of ROS by delaying the scavenging process, increasing enzymatic antioxidants, and accumulating osmotic substances [53]. When considering the climate data of the study site (Appendix A), it was found that the rainfall at mid-ripening (week 13) was 52.02% lower than the previous month, while the temperature remained the same. Therefore, BTH may have strengthened the accumulation of ROS in grapevines under drought stress, which led to the conversion of more linolenic and linoleic acids into malondialdehyde. Consequently, it decreased the content of linolenic and linoleic acids. This may be the main reason for the significant reduction in linoleic and linolenic acid contents at mid-ripening (week 13) of BTH-treated grapes.

### 3.3. Effect of BTH Treatment on the Activities of Key Enzymes of LOX Pathway

LOX activity reflects the grape maturity and degradation of membrane lipids. LOX activity increased and then decreased during grape ripening, with the greatest increase observed during veraison. The LOX activity of BTH-treated and CK grapes increased by 50.32% and 62.03%, respectively, at end-veraison (week 11) compared with mid-veraison (week 9). BTH-treated and CK grapes reached their peaks at end-veraison (week 11) and the mid-ripening stage (week 13), respectively (Figure 3A). Deytieux et al. [54] found that LOX activity was closely related to the external environment. They also observed variations in the distribution of specific proteins in grape skin at different stages of ripening. During veraison, there is an overexpression of stress and defense-related proteins, which is consistent with the increase in LOX activity. The peak of LOX activity in grapes treated with BTH was advanced. This could be attributed to the enhanced stress and disease resistance of grapevines due to BTH treatment, which activates the response mechanism against diseases.

HPL catalyzes the cleavage of lipohydroperoxides to C6 or C9 aldehydes and is a member of the *CYP74* gene family within the cytochrome P450 protein family. The activity of HPL increased and then decreased during grape ripening. It reached its peak at end-veraison (week 11) and decreased during ripening (week 15) to a level comparable to mid-veraison grapes (week 9) during grape ripening, which is in agreement with the findings of OuYang et al. [55]. The HPL activity of CK and BTH-treated grapes increased by 116.32% and 73.71% from mid-veraison to end-veraison, respectively. The HPL activity of BTH-treated grapes was significantly reduced by 15.80% compared to the CK at end-veraison (week 11) (Figure 3B). Reports have shown that, in the potato [56], *Arabidopsis* [57], and tomato [58], plant HPL activity is specific to tissue development. In particular, HPL expression is higher in fruits at the softening stage than in other periods and higher in flowers than in other organs. Jasmonic acid (JA) and its derivatives are produced via the allene oxide synthase (AOS) branch of the LOX pathway. The initial precursor for JA synthesis is linolenic acid, which enters the cytoplasm and is oxidized by LOX to 13-hydroperoxy-octadecatrienoic acid. This compound is subsequently catalyzed to 12-oxophytodienoic acid through the action of AOS and allene oxide cyclase. 12-oxophytodienoic acid is converted into JA through the action of 12-oxo-phytodienoic acid reductase and three rounds of β-oxidation. Notably, HPL and AOS exhibit antagonistic behavior due to their membership in the same cytochrome P450 family in plant skin cells. Thus, BTH treatment in grapes results in a decrease in HPL activity due to the overactivation of the JA-generating AOS branch.

ADH is a zinc-containing enzyme in plants that reduces C6 aldehydes to their corresponding C6 alcohols, with the assistance of the nicotinamide adenine dinucleotide and nicotinamide adenine dinucleotide phosphate. ADH activity is related to the plant response to environmental stimuli and cultivation practices. In this study, the activity of ADH showed an initial increase, followed by a decrease, reaching its highest level at end-veraison (week 11) in BTH-treated grapes and at mid-ripening (week 13) in CK grapes. The activity of ADH in BTH-treated grapes was significantly higher by 46.44% and 13.48% compared with the CK at the mid-veraison and end-veraison, respectively (Figure 3C). This would accelerate the conversion of C6 aldehydes to C6 alcohols in veraison grapes, leading to a decrease in C6 aldehydes in ripe grapes treated with BTH. Notably, ADH activity decreased slightly in BTH-treated grapes at weeks 13 and 15, unlike the significant increase at weeks 9 and 11 (Figure 3C). It has been demonstrated that the decrease in ADH activity can explain the decrease in hexanol, E-2-hexenal, and E-3-hexenol during tomato refrigerated storage. *ADH1* was found in seeds and young seedlings, and *ADH2* accumulates in tomato fruit during ripening [59]. It can be hypothesized that the erratic trend of the ADH activity in our study is due to the tissue specificity of grapevine ADH and the interaction of E-2-hexenal.

AAT transfers acyl groups from acyl-CoA to alcohol substrates to form esters, and C6 alcohols are further converted by AAT to hexyl acetate, Z-3-hexenyl acetate, and E-2-hexenyl acetate. AAT activity fluctuated during grape ripening, peaking at mid-ripening and then decreasing to levels comparable to that of mid-veraison (week 9). BTH treatment did not change the trend of AAT activity during grape ripening but significantly increased AAT activity at mid-ripening by 30.99% compared to the CK (Figure 3D). Defilippi et al. [60] described in their study that the regulation of *MdAAT2* by SA is mediated by the induction of endogenous ethylene synthesis and the activation of the ethylene signaling pathway. They also observed that SA significantly suppressed the expression level of ethylene-producing genes (*ACO1* and *ACS2*) [61]; inhibited the conversion of 1-aminocyclopropane-1-carboxylic acid to ethylene; and decreased the respiration rate depending on the inhibition of ethylene synthesis and stomata closure [61]. Therefore, BTH may inhibit ethylene synthesis by affecting the rate of grape cell respiration and the activities of two ethylene synthases. Consequently, this attenuates the regulation of AAT activity by ethylene and enhances AAT activity, since the regulation of AAT by BTH is dependent on influencing the respiration rate and ethylene synthesis, which are influenced by a variety of factors (variety and developmental stage, etc.) during plant growth [60]. This may lead to a weakening of the effect of BTH on AAT, and AAT thus exhibits an unstable tendency.

### 3.4. Effect of BTH Treatment on the Concentration of C6 Aldehydes, Alcohols, and Esters in Cabernet Gernischt Grapes

C6 aldehydes, alcohols, and esters can be synthesized from linoleic and linolenic acids through the catalytic action of LOX, HPL, ADH, and AAT enzymes. C6 aldehydes, alcohols, and esters in grapes exhibit herbal, citrus, and green leafy aromas. Hexanal, hexanol, E-2-hexenal, and E-2-hexenol are the direct precursors of hexyl acetate, an important compound that contributes to the aroma of grapes and wine during fermentation. During the ripening of grapes, a total of sixteen C6 compounds were detected, which included bound and free aldehydes (three), alcohols (three), and esters (two) (Figure 4). The same classes of free and bound C6 compounds were detected in both BTH-treated and CK grapes, indicating that the classes of C6 compounds remained unchanged following BTH treatment. The levels of free C6 aldehydes and C6 alcohols showed an increasing then decreasing trend during grape ripening. C6 alcohols increased sharply at veraison, while both C6 aldehydes and alcohols decreased at ripe grapes, with the trends corresponding to HPL and ADH activities, respectively. C6 esters showed a continuous decline during grape ripening, with the greatest decline in the Z-3-hexenyl acetate concentration (Figure 4). Furthermore, the concentration of C6 aldehydes and alcohols in ripe grapes treated with BTH was significantly lower compared to untreated grapes (CK), while the concentration of C6 esters was significantly higher in the BTH-treated grapes compared to the CK.

The Importance of wine aroma on the sensory profile should not be overlooked, as it is a key factor that leads to the consumer’s choice of wine [62]. However, HPL activity was significantly lower in BTH-treated grapes than in the CK at end-veraison stage, and this difference in enzyme activities resulted in variations in the concentration of C6 aldehydes and C6 alcohols. A significant decrease in C6 compounds was also obtained by silencing HPL in potatoes [56] The findings suggests that BTH inhibits the conversion of precursor 13-hydroperoxy-octadecatrienoic and 13-hydroperoxy-octadecadienoicnoic acids to C6 aldehydes and C6 aldehydes to C6 alcohols by decreasing the activities of HPL and ADH and promotes the conversion of C6 alcohols to C6 esters by increasing AAT activity. The concentration of bound C6 compounds in Cabernet Gernischt grapes was much lower than that of the free form, indicating that the majority of C6 compounds in these grapes exist in their free form. The developmental patterns of C6 aldehydes and alcohols, both in bound and free forms, exhibit similar trends. However, C6 esters demonstrate contrasting patterns. Interestingly, the significant effects of BTH treatments on bound C6 aldehydes and C6 alcohols were mainly observed before mid-veraison (weeks 7–11), whereas the significant effects on bound C6 esters were mainly observed after veraison (weeks 13–15). This suggests a lag in the impact of BTH treatment on C6 compounds, as elaborated in the OPLS-DA analysis score plot (Figure 5A). BTH probably increased the accumulation of linoleic and linolenic acid and metabolic substrates during the end-swelling and mid-veraison stages (weeks 7 and 9, respectively). This, in turn, significantly affected the activities of LOX, HPL, and ADH enzymes during end-veraison (week 11), consequently impacting the concentration of C6 compounds in mid-ripening and ripe grapes. Previous studies have shown that BTH significantly affects the concentration of C6 compounds and that the increase in C6 aldehydes may be due to the alteration of fatty acid formation pathways by BTH [22,31]. However, in our study, linolenic and linoleic acids were significantly reduced, and BTH appeared to continuously promote the whole metabolic process as the grapes progressed through the grape ripening process. The increase in C6 ester concentration in BTH-treated grapes is strong evidence for this promotion [22].

#### 3.4.1. C6 Aldehydes

In BTH-treated and CK grapes, the most abundant C6 aldehyde was free E-2-hexenal, with concentrations ranging from 1852.08 to 3779.17 μg/L. Hexanal was the second most abundant, while Z-3-hexenal was relatively the least abundant (Appendix A). Free E-2-hexenal increased sharply before veraison, peaked at end-veraison (week 11), and declined in ripe grapes (Figure 4). Nevertheless, BTH treatment did not change the trend of accumulation of free E-2-hexenal. Specifically, there was a notable 31.83% increase in end-swelling BTH-treated grapes compared to the CK, while ripe grapes (week 15) treated with BTH experienced a significant decrease of 15.80%. The enzyme 3Z,2E-enal isomerase plays a crucial role in converting Z-3-hexenal to E-2-hexenal and is active in grape berries [1]. In this study, a decrease in the concentration of free E-2-hexenal was observed in BTH-treated ripe grapes, which was speculated to be due to the inhibition of 3Z,2E-enal isomerase activity by BTH, which reduced the cleavage of Z-3-hexenal to E-2-hexenal. The concentration of bound E-2-hexenal during grape ripening varied from 3.23 to 95.60 μg/L. Ripe grapes treated with BTH showed a significant reduction of 56.68% in bound E-2-hexenal compared with CK (Appendix A). The concentration of free hexanal initially increased and subsequently decreased. BTH treatment resulted in an increase in the concentration of free hexanal during the end-swelling and mid-veraison stages (weeks 7 and 9, respectively). However, during the end-veraison to ripe grape stages (weeks 11–15) the concentration of free hexanal decreased with BTH treatment. Notably, the concentration of free hexanal in BTH-treated mid-ripening grapes (week 13) was significantly reduced by 18.91%. This could be attributed to the inhibitory effect of BTH on 13-HPL activity, resulting in a decrease in the conversion of 13-hydroperoxy-octadecadienoicnoic acid to hexanal. The trends of bound and free hexanal were basically the same. Due to the large proportion of E-2-hexenal and hexanal concentration in C6 compounds, changes in the levels of E-2-hexenal and hexanal directly affect the overall aroma of grapes, and the reductions in E-2-hexenal and hexanal directly diminish the green, apple, and fruity aroma of grapes.

#### 3.4.2. C6 Alcohols

C6 alcohols, such as E-2-hexenol, Z-3-hexenol, and hexanol, are direct products of ADH and are the immediate substrates for the formation of C6 esters in grapes. The ratio of Z-3-hexenol to E-2-hexenal can be used to determine the maturity level of grapes from an aroma point of view, with a decreasing trend observed as grapes ripen. Except for Z-3-hexenol, both E-2-hexenol and hexanol tended to increase during grape ripening. Salinas et al. [63] also reported that the Z-3-hexenol concentration in Monastrell grapes declined two weeks after veraison. BTH treatment resulted in a significant increase in AAT activity in mid-ripening grapes (Figure 3D). This increase in AAT activity led to the conversion of a higher amount of E-2-hexenol and hexanol into E-2-hexenyl acetate and hexyl acetate. Consequently, there were significant reductions of 36.74% and 33.44% in the levels of free E-2-hexenol and hexanol, respectively, in ripe BTH-treated grapes compared to the CK (Appendix A). Wu et al. [1] also found that AAT activity and C6 esters showed a very similar trend of elevation in Shine Muscat grapes near harvest, which supports our speculation. From the point of aroma maturation, BTH was found to promote aroma changes in Cabernet Gernischt grapes. However, in terms of TSSs and reducing sugar content, BTH was found to delay the technological ripening of these grapes. This indicates that aroma changes and the optimal technical ripening of grape berries may not occur at the same time.

#### 3.4.3. C6 Esters

A total of two C6 esters, specifically bound and free hexyl acetate and Z-3-hexenyl acetate, were detected at low levels during grape ripening. Free Z-3-hexenyl acetate accumulated mainly in the early stages of the ripening process. The continuous decrease in free Z-3-hexenyl acetate during grape ripening observed in our study may be attributed to the progressive conversion of its free to bound form, as bound Z-3-hexenyl acetate increased during grape ripening. Z-3-Hexenyl acetate can be used for the identification of grapes at the fruit-set and pre-veraison stages [35]. In comparison, the evolutionary pattern of hexyl acetate is less stable than that of Z-3-hexenyl acetate. The hexyl acetate concentration generally showed a decreasing trend and then an increasing trend during grape ripening. BTH treatment significantly enhanced the free hexyl acetate concentration in veraison and ripe grapes, especially in ripe grapes, with a significant increase of 192.71%. Similarly, the free Z-3-hexenyl acetate concentration was significantly increased by 214.32% in ripe BTH-treated grapes compared with the CK (Appendix A). This could be explained by the decrease in the concentration of bound Z-3-hexenol, the enhancement of AAT activity, and the increase in the conversion of Z-3-hexenol and hexanol to their corresponding esters. In this study, the high concentration of free hexyl acetate corresponds to the high activity of AAT during grape ripening, which is similar to the results of the study described by Gong et al. [33] using BTH.

#### 3.4.4. OPLS-DA Analysis of C6 Compounds in Cabernet Gernischt Grapes

To visualize the effect of BTH on the accumulation of C6 compounds, OPLS-DA analysis was performed using the concentration of free and bound C6 compounds identified during grape ripening as the variables. Figure 5A shows a score plot where the horizontal coordinates represent inter-group differences and the vertical coordinates represent intra-group differences in scores. BTH-treated samples were distributed on the negative x-axis, while CK samples were distributed on the positive x-axis, suggesting that BTH treatments had a substantial effect on C6 compounds. The distribution of sample points followed a “V” shape, extending from the negative to the positive half-axis of the y-axis, and the separation of sample points between BTH-treated and CK grapes was gradually evident from mid-veraison to grape ripening (weeks 9–15). This suggests that the impact of BTH treatment on C6 compounds became increasingly prominent as the grapes matured. Furthermore, the disparity in the type and concentration of C6 compounds between BTH-treated and CK grapes was the most significant in ripe grapes. The VIP scores plot (Figure 5B) further illustrated that BTH treatment resulted in a significant increase in the levels of bound Z-3-hexenyl acetate, hexyl acetate, and free hexyl acetate, while also leading to a significant decrease in the levels of bound Z-3-hexenol and hexanol in grapes.

### 3.5. Pearson Correlation Analysis of Linoleic Acid, Linolenic Acid, Enzyme Activity, and C6 Compounds

To investigate the effects of BTH treatment on the correlation between substrates, activities of key enzymes, and free and bound C6 compounds, Pearson correlation analysis was carried out on the relevant indexes during grape ripening. The correlation analysis conducted on CK grapes revealed that the nodes corresponding to linolenic acid, ADH, and LOX exhibited a higher intensity of red coloration (Figure 6A), indicating a stronger correlation between these nodes and the other fractions under investigation. Specifically, linolenic acid, free C6 aldehydes, C6 alcohols, total free C6 compounds, LOX, ADH, and AAT showed highly significant positive correlations. This suggests that linolenic acid derived from grapes has a positive effect on the accumulation of C6 alcohols and the activity of enzymes involved in their production (see Appendix A). Furthermore, the results revealed a strong positive correlation between ADH and C6 alcohols, as well as a strong negative correlation between ADH and free C6 esters. These findings suggest that ADH plays a role in the regulation of C6 alcohols and free C6 esters, promoting the former and inhibiting the latter. LOX also showed significant positive correlations with linolenic acid, free C6 aldehydes, C6 alcohols, and total free C6 compounds (Appendix A), indicating that LOX and ADH are key pacemaker regulatory enzymes in the anabolic pathway of C6 compounds, which is consistent with the results reported by Wu et al. [1].

Correlation analysis of BTH-treated grapes revealed that linolenic acid, LOX, HPL, and ADH exhibited the most significant correlations with the other fractions, as depicted in Figure 6B. BTH treatment resulted in an increased correlation between HPL and the other fractions, particularly a positive correlation with free and bound C6 aldehydes, as well as bound C6 compounds. This suggests that the BTH treatment has a positive regulatory effect on these fractions through HPL. Moreover, the inhibition of HPL in tobacco and *Arabidopsis* also yielded significant reductions in C6 compounds, with a strong positive correlation between the expression of the gene encoding the HPL enzyme and total C6 compounds [64]. In contrast, free C6 esters were negatively correlated with linolenic acid, LOX, HPL, and ADH, suggesting that free C6 esters in grapes are indirectly influenced by these parameters, in addition to being directly influenced by AAT in a positive manner. BTH treatment enhanced the positive correlation between LOX and bound C6 aldehydes, C6 esters, and bound C6 compounds. Additionally, it was observed that BTH treatment decreased the correlation between LOX and linolenic acid. These findings suggest that BTH treatment has the potential to enhance the relationship between bound C6 compounds and other fractions by regulating LOX activity. Furthermore, BTH treatment can significantly increase the concentration of bound C6 compounds and facilitate the degradation of linolenic acid by LOX.

The concentration of compounds in the aroma substances metabolism depends on the activity of key enzymes, substrate specificity, and substrate availability. The limiting factors of metabolism can be clarified by studying the correlation between the substrate concentration, enzyme activity, and product concentration. The observed variations in linoleic and linolenic acids in our investigation did not align with the trends observed for E-2-hexenal and hexanal. This indicates that the levels of linoleic and linolenic acids are unlikely to be the determining factor in the LOX pathway. Instead, it is more likely that the activity of LOX and ADH enzymes plays a significant role. The trends observed in AAT activity and hexyl acetate concentration were similar, suggesting that AAT is a limiting factor in determining the formation of C6 esters in grapes, which is similar to the results reported by Zhao et al. [65] but different from the findings of Wu et al. [1]. The disparities between the studies could be because the change in enzyme activity is influenced not only by changes in mRNA expression but also by the transcriptional regulation and cellular metabolism. Other mechanisms of BTH treatment are yet to be confirmed in future research.

The findings of this study are limited to physicochemical parameters, enzyme activities, and aroma substance concentrations, which are considered adequate to determine the effect of BTH on C6 compounds. However, consideration of a number of factors in future studies would provide more information for determining the mechanism by which BTH affects C6 compounds, such as the consistency of results between vintages, determination of key gene expression levels (*VvLOXA*, *VvHPL1*, *VvADH2*, *VvAAT*, etc.), and differences between varieties.

## 4. Conclusions

BTH treatment at the pre-swelling stage significantly increased the 100-berry weight, skin/berry ratio, pH, total phenolic content, and total flavonoid content in ripe Cabernet Gernischt grapes. Additionally, BTH treatment reduced the content of reducing sugars, TSSs, and TA. BTH treatment also delayed the technological ripening of grapes by retarding sugar accumulation. Furthermore, BTH promoted the conversion of C6 aldehydes to C6 alcohols and C6 alcohols to C6 esters by increasing the activities of LOX, ADH, and AAT. As a result, the content of hexyl acetate and Z-3-hexenyl acetate in ripe grapes increased significantly, while the concentration of free E-2-hexenal and C6 compounds decreased. BTH treatment also advanced the changes in the grape aroma by promoting ester production. OPLS-DA and correlation analysis showed that the accumulation of C6 compounds varied greatly after BTH treatment and that the effect of BTH treatment on C6 compounds was progressively strengthened with an increasing grape maturity. BTH treatment had a greater effect on the accumulation of bound Z-3-hexenyl acetate, hexyl acetate, Z-3-hexenol, hexanol, and free hexyl acetate. Also, the activities of LOX and ADH, which are the limiting factors of the LOX pathway, positively regulated the synthesis and accumulation of C6 esters.

## Figures and Tables

**Figure 1 foods-12-03710-f001:**
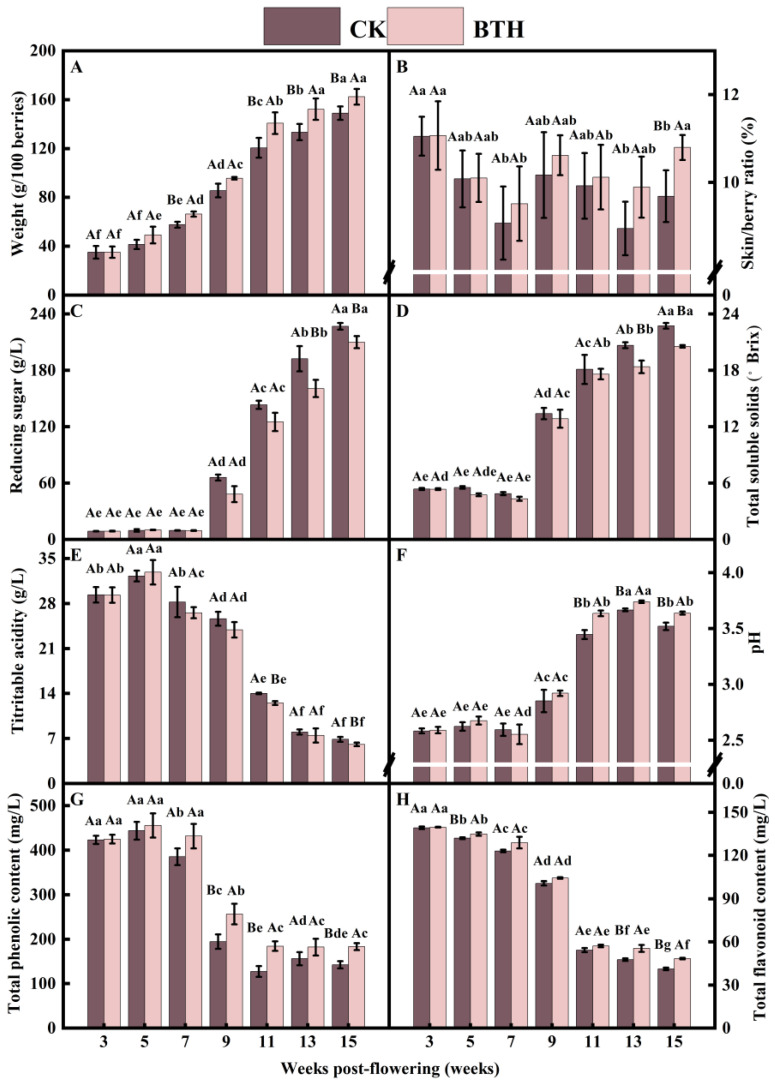
Physicochemical parameters of BTH-treated (BTH) and control (CK) Cabernet Gernischt grapes. Among them are the 100-berry weight (**A**), skin/berry ratio (**B**), reducing sugars (**C**), total soluble solids (**D**), titratable acidity (**E**), pH (**F**), total phenolics, (**G**) and total flavonoids (**H**). Data are expressed as means (*n* = 3). Error bars represent standard deviation. Different uppercase letters indicate significant differences between BTH and CK (*p* < 0.05), while different lowercase letters indicate significant differences within CK or BTH between different weeks of post-flowering (*p* < 0.05).

**Figure 2 foods-12-03710-f002:**
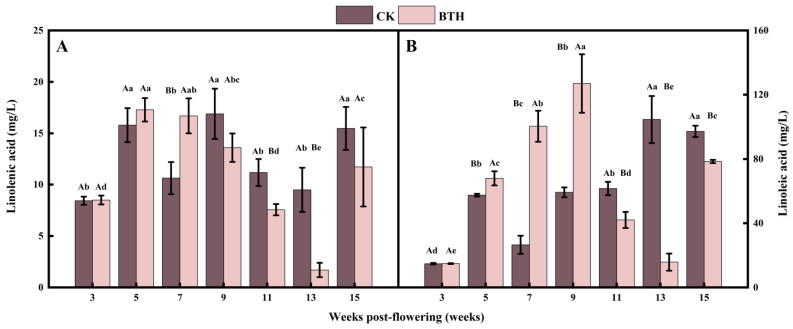
Linolenic acid (**A**) and linoleic acid (**B**) of BTH-treated (BTH) and control (CK) Cabernet Gernischt grapes. Data are expressed as means (*n* = 3). Error bars represent standard deviation. Different uppercase letters indicate significant differences between BTH and CK (*p* < 0.05), while different lowercase letters indicate significant differences within CK or BTH between different weeks of post-flowering (*p* < 0.05).

**Figure 3 foods-12-03710-f003:**
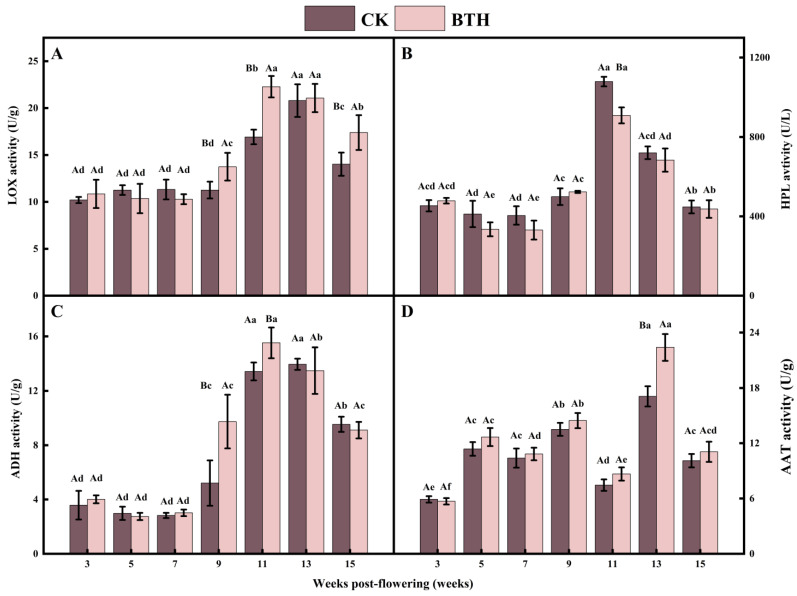
LOX (**A**), HPL (**B**), ADH (**C**), and AAT (**D**) activities of BTH-treated (BTH) and control (CK) Cabernet Gernischt grapes. Data are expressed as means (*n* = 3). Error bars represent standard deviation. Different uppercase letters indicate significant differences between BTH and CK (*p* < 0.05), while different lowercase letters indicate significant differences within CK or BTH between different weeks of post-flowering (*p* < 0.05).

**Figure 4 foods-12-03710-f004:**
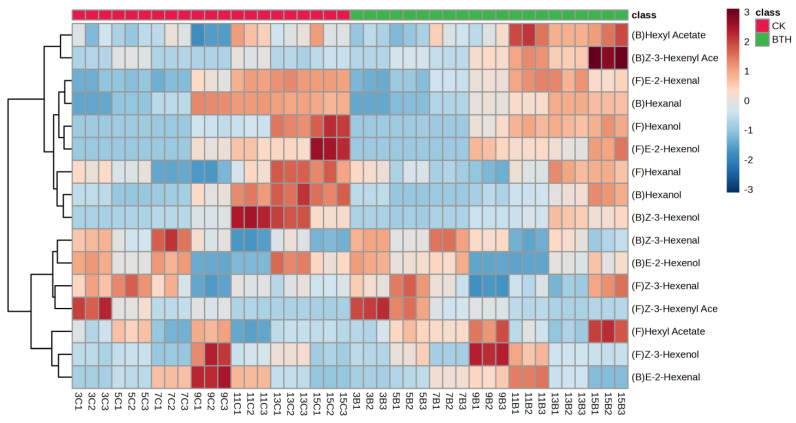
Hierarchical heatmap of C6 compounds in control (CK) and BTH-treated (BTH) Cabernet Gernischt grapes during grape ripening. The color scale shows that compounds with high concentrations range from white to dark red, while compounds with low concentrations range from white to dark blue. “(F)” and “(B)” indicate free and bound aromas, respectively. Each line consists of three measurements of the same sample to achieve minimum error. The first digit of the sample name indicates the number of weeks post-flowering. In the second position, “B” indicates BTH-treated grapes and “C” indicates CK grapes. For example, “3C1” represents the first parallel measurement of CK grapes at 3 weeks post-flowering.

**Figure 5 foods-12-03710-f005:**
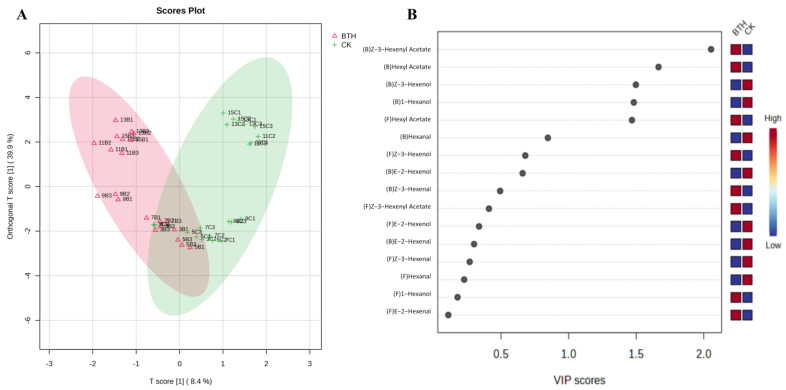
OPLS-DA analysis scores plot (**A**) and VIP scores plot (**B**) of C6 compounds in control (CK) and BTH-treated (BTH) Cabernet Gernischt grapes during grape ripening. “(F)” and “(B)” indicate free and bound aromas, respectively. Sample points with suffixes 1, 2, and 3 are three measurements of the same sample in order to attain minimum error. The first digit of the sample name indicates the number of weeks after flowering. In the second position, “B” indicates BTH-treated grapes and “C” indicates CK grapes. For example, “3C1” represents the first parallel measurement of CK grapes at 3 weeks post-flowering.

**Figure 6 foods-12-03710-f006:**
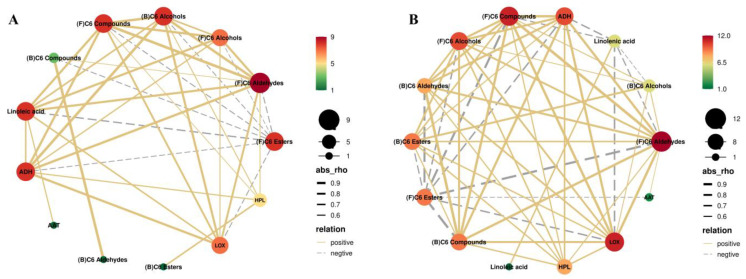
Pearson correlation analysis of linoleic acid, linolenic acid, enzyme activity, and C6 compounds in CK (**A**) and BTH (**B**) grapes during grape ripening. “(F)” and “(B)” indicate free and bound aromas.

## Data Availability

Data are contained within the article and the Appendix A.

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
