# Peer review of "Influence of Benzothiadiazole on the Accumulation and Metabolism of C6 Compounds in Cabernet Gernischt Grapes (Vitis vinifera L.)"

_foods, 2023, doi:10.3390/foods12193710_

Round 1
Reviewer 1 Report
This study is of scientific interest in the field of grape crop and enology. The authors studied the impact of benzothiadiazole (BTH) on the physicochemical properties and C6 compounds of Cabernet Gernischt grapes during ripening. The work is properly written and presented. The experimental design is appropriate, the methods are adequately described with a proper statistical analysis, the results are clearly presented and discussed, with promising perspectives. Thus, I consider that this manuscript can be published in Foods after correct some minor observations.
Minor revisions:
Line 96: Please, correct the first sentence of the paragraph accordingly.
Line 256: Fig. 1- It seems that the lower-case letter over the bars compare only between the week post-flowering inside a treatment (CK or BTH). Please clarify.
Line 300: Please correct "as a result".
Line 313: Please, correct with capital letter "Phosphofructokinase"
Line 408: Figure 2A- Please, revise the significant differences (capital letters) BTH-treated grapes and CK grapes, specifically in weeks 7 and 13. It seems that there are significant differences.
Line 498: Figure3- Please, revise the capital letters for statistical analysis in 3C and 3D in 11 and 15 weeks.
Line 524: It is not consistent with results of the figure 3 for 15 weeks (or revise the statistical analysis).
Lines 554-556: …"the activity of this enzyme was inhibited by BTH"… to say this, you have to prove it; if not, you have to hypothesize that it could be the cause. Please, rewrite.
Lines 579-580: With respect to the sentence "BTH treatment resulted in a significant increase in AAT activity in ripe grapes (Table S3).", you should change the statement because you are not expressing the direct results of the AAT activity of this enzyme in Table S3; with these results you are attributing the aroma compounds formation to the AAT enzyme action. Please, correct accordingly.
Line 591: Please, add a coma after Z-3-hexenyl acetate.
Lines 592-594: In this sentence, did you mean that this behavior is due to ripening process or due to BTH treatment?
Lines 597-598: ...with respect to what? the time of ripening? Free or bound C6 compounds? Please, clarify this sentence and the first ones of this paragraph.
Line 601: Specify if these are free or bound C6 compounds.
Lines 619-620: Why do you say from this graph that BTH treatment had a greater effect on C6 compounds?
Line 641: Add a period at the end of the sentence.
Reviewer 2 Report
Dear Authors,
This is very interesting study related to investigation of application of BTH in the improvement of grape quality with a highlight of accumulation and metabolism of C6 compounds.
Find my suggestions below.
In the abstract is more than 200 words. Make abstract according to instruction to authors.
In the line 53 start sentence with the full word and put abbreviation in brackets. Use abbreviation further in manuscript.
In the line 48 highlight that grape maturity stage and phytosanitary condition of grape significantly affect on the content of C6 compounds in grape. Kindly consider to cite Fermentation 9(7), (2023), 695.
Change the title of subsection 2.1. to Plant material and study site
In the subsection 2.1.highlight climate conditions in this part of China during the year and how it influenced on the grape crop.
Did you treated vineyard with the any chemicals for the protection? Highlight it in the subsection 2.1.
Which fertilizer was used in vineyard? Highlight it in the subsection 2.1.
Highlight in the subsection 2.1. how was phytosantiry state of grape. Was it affected by grey mold?
Add new subsection 2.2. entitled Reagents and chemicals.
Which spectra were used for the quantification in the subsection 2.7.? Did gas chromatograph was equipped with mass detector? Highlight it in subsection 2.7. it is very important.
In the line 527 highlight the importance of volatile compounds of wine as a key factors for the wine selection by consumers. Kindly consider to cite J. Serb. Chem. Soc. Vol. 88 No. 1 (2022) 11-23.
In the subsection 2.8. highlight which parameters were compared in one-way and which in two-way ANOVA analysis.
Wish you all the best in the future work,
Reviewer 3 Report
The present manuscript is well written and well presented. Small changes and corrections should be done, for example the punctuation marks need to be checked. In general terms, the authors have done a very good job. The use of elicitors in grape production is of great interest and several researchers have already studied them. BTH is one of them and the present manuscript contributes positively with new information related to the C6 compounds formation on grapes.
I have two only comments. Firstly, the studies on the filed need to be done more than one harvest in order to check if the results are constant.
The second one is that the linoleic and linolenic acid concentrations as also the ADH and AAT activities didn't have a clear influence of the BTH application (positive or negative). It is more an erratic trend. Do the authors have an answer on that?
Round 2
Reviewer 2 Report
Dear Authors,
Thank you very much for revised version of manuscript and detailed explanations. Also you were precisely showed in the cover letter all improvements of your manuscript. It is fine for me.
Wish you all the best in the future work,